# A data integration approach unveils a transcriptional signature of type 2 diabetes progression in rat and human islets

**Shenghao Cao**[1,2], **Linting Wang**[1,2], **Yance Feng**[1,2], **Xiao-ding Peng**[3], **Lei M. Li**[1,2]*

**1** National Center of Mathematics and Interdisciplinary Sciences, Academy of Mathematics and Systems Science, Chinese Academy of Sciences, Beijing, China, **2** University of the Chinese Academy of Sciences, Beijing, China, **3** Department of Biochemistry and Molecular Genetics, The University of Illinois at Chicago, Chicago, Illinois, United States of America

☯ These authors contributed equally to this work.
* lilei@amss.ac.cn

**Data Availability Statement:** All data including normalized expression profiles and scripts of data integration analysis, as well as the scripts used to generate graphs and figures, are deposited as

## Abstract

Pancreatic islet failure is a key characteristic of type 2 diabetes besides insulin resistance. To get molecular insights into the pathology of islets in type 2 diabetes, we developed a computational approach to integrating expression profiles of Goto-Kakizaki and Wistar rat islets from a designed experiment with those of the human islets from an observational study. A principal gene-eigenvector in the expression profiles characterized by up-regulated angiogenesis and down-regulated oxidative phosphorylation was identified conserved across the two species. In the case of Goto-Kakizaki versus Wistar islets, such alteration in gene expression can be verified directly by the treatment-control tests over time, and corresponds to the alteration of α/β-cell distribution obtained by quantifying the islet micrographs. Furthermore, the correspondence between the dual sample- and gene-eigenvectors unveils more delicate structures. In the case of rats, the up- and down-trend of insulin mRNA levels before and after week 8 correspond respectively to the top two principal eigenvectors. In the case of human, the top two principal eigenvectors correspond respectively to the late and early stages of diabetes. According to the aggregated expression signature, a large portion of genes involved in the hypoxia-inducible factor signaling pathway, which activates transcription of angiogenesis, were significantly up-regulated. Furthermore, top-ranked anti-angiogenic genes *THBS1* and *PEDF* indicate the existence of a counteractive mechanism that is in line with thickened and fragmented capillaries found in the deteriorated islets. Overall, the integrative analysis unravels the principal transcriptional alterations underlying the islet deterioration of morphology and insulin secretion along type 2 diabetes progression.

## Introduction

Pancreatic islet failure is a key characteristic of type 2 diabetes (T2D) besides insulin resistance [1]. Clarifying the molecular mechanisms underlying the islet failure can aid in the prevention and treatment of T2D. With the available whole-genome transcriptome, three questions arise

supplementary data on https://codeocean.com with the DOI: https://doi.org/10.24433/CO.1261622.v3. The programs and intermediate results of normalization are deposited on https://zenodo.org with the DOI: https://doi.org/10.5281/zenodo.7658587 for rat islets and https://doi.org/10.5281/zenodo.7659808 for human islets.

**Funding:** SC, LW, YF and LML acknowledges the funding from the National Key Research and Development Program of China (https://chinainnovationfunding.eu/national-key-rdprogrammes/) through grant number 2022YFA1004801, the National Natural Science Foundation of China (https://www.nsfc.gov.cn/) through grant numbers 11871462, 32170679, and 91530105, the National Center for Mathematics and Interdisciplinary Sciences, Chinese Academy of Sciences, and the Key Laboratory of Systems and Control, Chinese Academy of Sciences. The funders had no role in study design, data collection and analysis, decision to publish, or preparation of the manuscript.

**Competing interests:** The authors have declared that no competing interests exist.

naturally. First, is there a conserved transcriptional signature that characterizes islet deterioration in both T2D animal models and humans? Second, can the expression signature, if any, explains the insulin levels and observed morphological alterations in islets to some extent? Third, what factors or stress induce the regulation of expression signature? To answer these questions, this study developed a computational approach that integrates expression profiles of Goto-Kakizaki (GK) and Wistar (WST) rat islets from a designed experiment with those of the human islets from an observational study.

In an earlier report [2], we obtained the expression profiles of pancreas from the outbred mice fed with high-fat diet or regular chow at weeks 1, 9, and 18. The profiles were analyzed by the singular value decomposition (SVD) and dual eigen-analysis, in which the sample- and gene-eigenvectors corresponded respectively to the macro-factors and molecular biology. It turned out that the diet factor induced a clear contrast in the first principal sample-eigenvector; whereas in its dual gene-eigenvector, angiogenesis as downstream of the hypoxia-inducible factor (HIF) signaling pathway was remarkably up-regulated at the side corresponding to high-fat diet. In another study, up-regulation of some hypoxia-related genes was reported in islets of pre-diabetic male Zucker diabetic fatty (ZDF) rats [3]. However, whether the alteration of angiogenesis- and hypoxia-related gene expression is a signature in human islets remains unclear.

On the other hand, morphological alterations of islet blood vessels have indeed been observed in both T2D rodents and human. In ZDF rats, the islet microvasculature was reported to increase approximately twofold in young males [3]. Thickened endothelial cells were observed in 7-week-old ZDF rats.

In human T2D subjects, islet capillary morphology is abnormal [4,5] as well. Islets in pancreatic sections showed increased capillary density, thickened and fragmented capillaries as compared to non-diabetic controls. Particularly, intra-islet capillaries in T2D islets were frequently thicker as compared with those in control islets. Abnormal islet capillary morphology may contribute to a deterioration of β-cell/islet function [4,5].

Blood vessels not only provide metabolic sustenance, but also provide signals for islet development [6]. In the context of pancreas development, overexpression of VEGF-A (vascular endothelial growth factor-A) increased vascular endothelial cell number, which would impair islet morphogenesis with α- and β-cells being scattered and never coalesced into islets [7]. On the other hand, α- and β-cells of normal rodent islets have a non-random core-mantle organization [8], while those of diabetic islets showed a more mixing distribution [9–11]. The distribution of α/β-cells from center to periphery has been quantified [12]. However, the connection between alterations in islet vasculature and those in α/β-cell distribution during T2D progression remains unclear [8].

The study of pancreatic islets in humans with T2D faces challenges due to the limited accessibility of human islets and ethical issues. Instead, animal models are often used to investigate the diabetic mechanisms. In the GK rat model, the progression of T2D is spontaneous and more homogeneous over time among individuals [13]. The time-course expression profiles of pancreatic islets isolated from GK rats and WST rats [9] are very valuable for identifying the transcriptional signature of T2D progression. Moreover, we found that the spatial distributions of α- and β-cells of these samples can be drawn from those of glucagon and insulin immunohisto-staining available from the study [9]. Thereby we can investigate if the α/β-cell distribution is a key morphological marker along the T2D progression, and investigate its connection with transcriptional markers.

In contrast to rats, a human islet transcriptome dataset from 62 cadaver donors is available from an observational study [14]. The data not only show substantial variability in terms of their non-diabetic and T2D status at different stages, but also provide the physiological and

pathological indices of the donors such as $HbA_{1c}$ (glycosylated hemoglobin), BMI (body mass index), and age. These indices can be used to explore their association with molecular markers.

Identification of conserved molecular features shared by islets of an animal model and those of humans with T2D holds significant values. Thus, it is of great significance to align the islet transcriptomes of the two species and of the two different experiment designs in a justifiable way. In this study, we present such an integration framework that includes normalization, dual eigen-analysis, rank aggregation, data concatenation, and stability analysis by re-sampling. The results showed that islet transcriptional angiogenesis is a principal transcriptional alteration underlying islet deterioration in T2D progression.

## Materials and methods

The data and the scheme of the integration analysis are summarized in Fig 1.

### GK and WST rat islets

The expression data were downloaded from GEO under accession number GSE81811 [9]. The pancreatic islets were isolated from male GK rats and age-matched control WST rats at five consecutive time points: weeks 4, 6, 8, 16, and 24 of age. Their transcriptome data were obtained by RNA-seq using a multiplex analysis of polyA-linked sequences (MAPS) approach. Immunohistochemistry micrographs of GK and WST rat islet paraffin sections co-stained for insulin and glucagon at the five time points were available from the Supplementary Data of the original paper [9]. The data information is summarized in Fig 1A.

### Human islets

The expression data were downloaded from GEO under accession number GSE38642 [14]. The islets were from cadaver donors (54 non-diabetics and 9 diabetics). The mRNA abundances were measured by the Affymetrix GeneChip Human Gene 1.0 ST whole transcript. One non-diabetic sample did not pass the data quality test, and was excluded from further analysis. In addition to the information of age, gender, BMI and T2D status, the $HbA_{1c}$ levels of these individuals were collected as well. The data information is summarized in Fig 1A.

### Expression data normalization

The RNA-seq sequencing reads of rats were normalized by MUREN, a robust and multi-reference approach [15]. The goodness of normalization was evaluated by the principle we proposed [15,16]. In particular, the modes of densities of pairwise differences should be near zero after normalization. One such density of pairwise differences is shown in S1 Fig. The Affymetrix microarray data of human were normalized by sub-sub normalization [16,17] followed by the probe-treatment-reference summarization [18]. Guided by the same principle as above, we selected a set of references from raw samples to optimize the normalization results. One density of pairwise differences after normalization is shown in S2 Fig.

### SVD of gene expression profiles, empirical contrast and sum of squares

The scheme of SVD and dual eigen-analysis is shown in Fig 1B. Denote the log-scale gene expression profile from one species by a matrix $E = [E_{i,j}]$, where $i = 1, 2, \ldots, s$, and $j = 1, 2, \ldots, g$; $s$ is the number of samples, and $g$ is the total number of genes or probe sets. The SVD of $E$

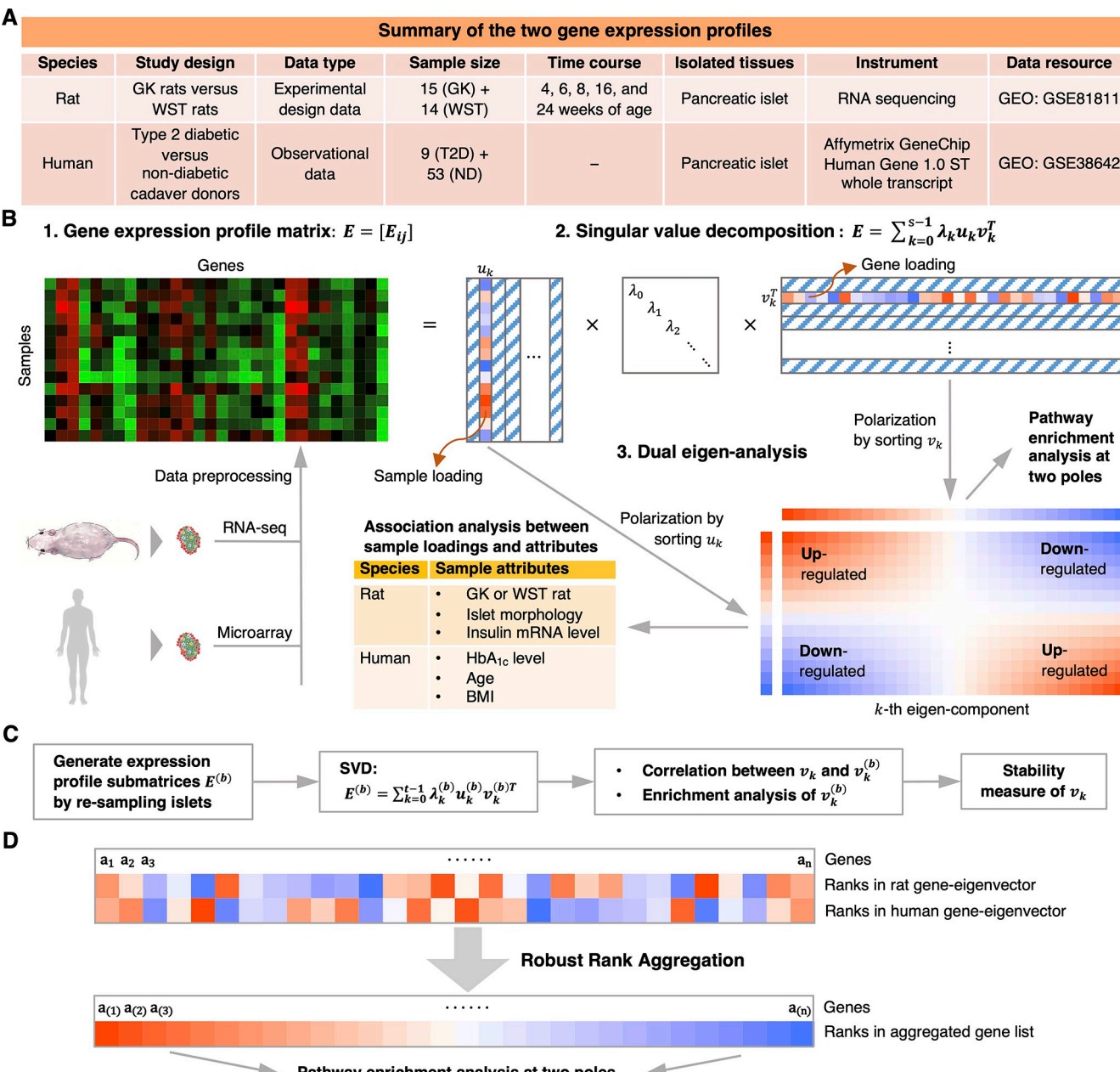

**Fig 1. Summary of the transcriptome data and scheme of the data integration analysis.** (**A**) The table outlines the experimental design and specifications of the two islet expression profiles used for dual eigen-analysis. GK, Goto-Kakizaki; WST, Wistar; T2D, type 2 diabetics; ND, non-diabetics. (**B**) The dual eigen-analysis procedure has several steps. First, the expression profile from one species is represented by a matrix in which the row and column indices correspond to samples and genes respectively. Second, calculate the SVD of the profile matrix; each eigen-component is comprised of a sample-eigenvector $u_k$ and its dual gene-eigenvector $v_k$, respectively corresponding to the macro- and micro-biology information; the singular value measures its weight in the profile. Third, polarize the sample- and gene-eigenvectors by sorting their loadings. Infer the association between the loadings of $u_k$ and sample attributes, and identify the biological pathways enriched at two poles of $v_k$. The pathways enriched at the positive pole (red) of $v_k$ are up-regulated in the samples at the positive pole (red) of $u_k$ whereas down-regulated in the samples at the negative pole (blue). Conversely, the pathways enriched at the negative pole (blue) of $v_k$ are down-regulated in the samples at the positive pole (red) of $u_k$ whereas up-regulated in the samples at the negative pole (blue). (**C**) Evaluate the stability of the gene-eigenvector by re-sampling islets. (**D**) Integrate the gene-eigenvectors from the two species into one gene list by Robust Rank Aggregation, and perform enrichment analysis on the aggregated gene list.

has the following representation:

$$E = \sum_{k=0}^{s-1} \lambda_k u_k v_k^T,$$

where $\lambda_k$ are positive and decreasing singular values; sample-eigenvectors $u_k$ and gene-eigenvectors $v_k$ are respectively of size $s$ and $g$; $\{u_k\}$ are mutually orthogonal and so are $\{v_k\}$; signs of the two vectors $u_k$ and $v_k$ can be reversed simultaneously without changing the product $u_k v_k{}^T$. The $k$-th principal eigen-component consists of $u_k$ and $v_k$, and its contribution to the total mRNAs is measured by the relative percentage of $\lambda_k$. The top eigen-component simply represents the baselines of genes and samples. For the sake of clarity, we set its index as zero when we refer to the $k$-th singular value or eigen-component.

The $k$-th principal sample-eigenvector $u_k$ is composed of weights of all samples, which are referred to as the sample loadings, whereas the dual gene-eigenvector $v_k$ is composed of weights of all genes, which are referred to as the gene loadings. Polarization of an eigenvector means the sorting of its loadings from low to high. The dual eigen-analysis [2,19,20] first polarizes $u_k$, and identifies the associations between loadings and sample attributes such as genetic factors and diabetic status; then polarizes $v_k$, and identifies the biological pathways enriched at its two poles. The activities of pathways enriched at the positive pole of $v_k$ are up-regulated in the samples with positive loadings, and are down-regulated in the samples with negative loadings. Conversely, the activities of pathways enriched at the negative pole of $v_k$ are down-regulated in the samples with positive loadings, and are up-regulated in the samples with negative loadings (part 3 in Fig 1B).

Denote the sample loadings of $u_k$ by $\{u_{k,1}, u_{k,2}, \ldots, u_{k,s}\}$, $k = 0, 1, 2, \ldots, s\text{-}1$. After normalization, the loadings of the baseline vector $\{u_{0,1}, u_{0,2}, \ldots, u_{0,s}\}$ are almost identical, with the coefficient of variations being 0.44% and 0.64% respectively in rat and human (S3 Fig). That is, $u_{0,i} \cong \frac{1}{\sqrt{s}}$, $i = 1, 2, \ldots, s$. Consequently, for $k \geq 1$, $\sum_{i=1}^{s} u_{k,i} \cong 0$ because of orthogonality, $\sum_{i=1}^{s} u_{0,i} u_{k,i} = 0$. Thus for $k \geq 1$, $\{u_{k,1}, u_{k,2}, \ldots, u_{k,s}\}$ is nearly a contrast [21] defined by data themselves. Therefore, we refer to it as an empirical contrast. After we extract the baseline, the profile becomes $[E_{i,j}^{(1)}] = E^{(1)} = E - \lambda_0 u_0 v_0^T$. We consider the contrast of the top principal sample-eigenvector $u_1$. Its induced squares for gene $j$ is $SS_j^{(1)} = (\sum_{i=1}^{s} u_{1,i} E_{i,j}^{(1)})^2 / \sum_{i=1}^{s} (u_{1,i})^2 = (\sum_{i=1}^{s} u_{1,i} E_{i,j}^{(1)})^2$. The total sum of squares is

$$SS_{constrast}^{(1)} = \sum_{j=1}^{g} (\sum_{i=1}^{s} u_{1,i} E_{i,j}^{(1)})^2 = \lambda_1^2.$$

According to the definition of SVD, $u_1$ is the maximizer of the following: $\max_{\|w\|=1} \|E^{(1)^T} w\|^2 = \max_{\|w\|=1} [\sum_{j=1}^{g} (\sum_{i=1}^{s} w_i E_{i,j}^{(1)})^2] = \lambda_1^2$, where $\|\cdot\|$ is the Euclidean norm [22]. In other words, $u_1$ is the contrast that explains the largest portion of the sum of squares. Similarly, we can explain $u_2$ by considering $[E_{i,j}^{(2)}] = E^{(2)} = E^{(1)} - \lambda_1 u_1 v_1^T$. The total sum of squares from all contrasts equals the square of the matrix Frobenius norm:

$$\sum_{k=1}^{s-1} \lambda_k^2 = \sum_{i=1}^{s} \sum_{j=1}^{g} [E_{i,j}^{(1)}]^2 = \|E^{(1)}\|_F^2,$$

where $\|\cdot\|_F$ represents the Frobenius norm.

The contrast in the sample loadings is an analog to that in a treatment and control design. If the expression profiles are from treatment and control samples as in the case of GK/WST rats, the identification of differentially expressed genes and the biological pathways they belong to is our primary concern. In one principal eigen-component of SVD, corresponding to the

contrast in its sample loadings is the gene-eigenvector that can be understood as the expression differentiation. We examine the genes at the two poles of the polarized gene-eigenvector. Furthermore, the meanings of these genes can be interpreted by the enrichment analysis. This correspondence between the sample- and gene-eigenvectors of SVD is what we termed as the dual eigen-analysis [2,19]. In the case of treatment versus control data, the results of dual eigen-analysis can be checked by the direct two-sample comparison. The advantages of dual eigen-analysis are demonstrated in S1 Text.

### Gene set enrichment by the Wilcoxon scoring method

The enrichment analysis of gene-eigenvector $v_k$ was based on the Wilcoxon scoring method that uses ranks of gene loadings. The method was proposed originally for the profiles of gene expression differences [23]. In this study, we considered gene sets from the following databases: Gene Ontology (GO) [24,25], Kyoto Encyclopedia of Genes and Genomes (KEGG) [26,27], and REACTOME [28,29].

### Comparison of GK/WST rat islet expression at each week

The differences of gene expression (after taking logarithm) between GK and WST rats were obtained after averaging the data of replicates. Then the enrichment analysis by Wilcoxon scoring was carried out for each gene subset.

### Evaluation of gene-eigenvector stability by re-sampling islets

The reliability of the data integration hinges on the stability of the principal eigenvectors in the singular value decomposition. To evaluate their stability, we generate from all the subjects a random sample $\{i_1^{(b)}, i_2^{(b)}, \ldots, i_t^{(b)}\}$ of a smaller size $t$, and then carry out singular value decomposition on the expression profiles of the sub-sample. In other words, we consider the SVD of the sub-matrix $E^{(b)} = [E_{ij}]$, where $i \in \{i_1^{(b)}, i_2^{(b)}, \ldots, i_t^{(b)}\}$, and $j = 1, 2, \ldots, g$.

$$E^{(b)} = \sum_{k=0}^{t-1} \lambda_k^{(b)} u_k^{(b)} v_k^{(b)T}.$$

Next, we compute the correlation coefficient of $v_k$ and $v_k^{(b)}$ for each pair of principal eigenvectors. Besides, we compare the enrichment results of $v_k$ and $v_k^{(b)}$. The re-sampling procedure is repeated for $b = 1, 2, \ldots, B$, where $B$ is a simulation parameter. The distribution of correlations between the sub-profiles and the complete profile offers a measure of stability (Fig 1C).

### Robust rank aggregation of gene-eigenvectors from two species

Other than comparing their gene set enrichment results, we directly integrated gene-eigenvectors from the two species into one gene list using the Robust Rank Aggregation method [30]. To evaluate the statistical significance of the aggregated ranks, a p-value was assigned for each gene in the aggregated gene list to describe how much better it was ranked than expected by chance. Throughout the article, the rank and p-value of a gene refer to the ones in the aggregated gene list, if not specified otherwise. We perform the enrichment analysis for the aggregated gene list as well (Fig 1D).

### Unified SVD of the concatenated expression profile matrix of two species

Other than aggregating the gene ranks from the SVDs of the two species, we can integrate the rat and human expression profiles of their homologous genes by directly concatenating them. Denote the homologous gene expression profiles of rat and human respectively by $_rE = [_rE_{i,j}]$

and $_hE = [_hE_{k,j}]$, where $j = 1, 2, \ldots, \bar{g}$, $i = 1, 2, \ldots, s_r$, $k = 1, 2, \ldots, s_h$, $s_r$ and $s_h$ are the rat and human sample sizes, and $\bar{g}$ is the number of homologous genes. To concatenate them properly, we need to remove their baselines. We first subtract from $_rE$ the baseline component $_rE^{(0)}$, determined by its SVD, and obtain $_rE^{(1)} = _rE - _rE^{(0)}$. Similarly, we obtain $_hE^{(1)}$. Then the two matrices with the same number of columns were concatenated into one single matrix $E = \begin{bmatrix} _rE^{(1)} \\ _hE^{(1)} \end{bmatrix}$. Next the unified SVD of the concatenated matrix $E$ is carried out,

$$E = \sum_{k=1}^{s_r+s_h} \lambda_k u_k v_k^T,$$

where the sample-eigenvector $u_k = \begin{bmatrix} _ru_k \\ _hu_k \end{bmatrix}$ can be partitioned into the component of rat and that of human. Finally, we applied the procedure of dual eigen-analysis as described in the subsection "SVD of gene expression profiles, empirical contrast and sum of squares". Furthermore, the variations from the two expression profiles are measured by their matrix or Frobenius norms. To equalize the variations, we opt to normalize $_rE^{(1)}$ and $_hE^{(1)}$ by dividing their Frobenius norms. Consequently, the normalized matrices are concatenated.

### Mouse islet micrographs

The islet micrographs from 6-month-old Wild-type (WT) and Akt1$^{+/-}$Akt2$^{-/-}$ (a type 2 diabetic model) male mice were from paraffin sections of pancreata co-stained for insulin and glucagon [31]. These micrographs were used to analyze the α/β-cell distribution and define the islet irregularity index.

### Quantification of the α- and β-cells arrangement and deterioration of islets

The morphology of GK rat islets altered from the core-mantle organization at weeks 4 and 6 into disorganized structures characterized by pronounced fibrosis separating strands of endocrine cells after week 8 [9]. We quantified the alteration of the α- and β-cells arrangement using the spatial distribution of glucagon and insulin, as exemplified by the islet micrographs of the above two WT and Akt1$^{+/-}$Akt2$^{-/-}$ mice [31]. In each islet micrograph, the rim of the islet was sketched by the convex hull of the insulin region, and the distances of glucagon pixels away from the islet rim were then defined. The kernel density estimation (KDE) of the distances roughly represented the spatial distribution of α- and β-cells.

To evaluate the deterioration of islets, we defined a single-valued morphometric index by the trimmed mean of the distances of glucagon pixels away from the islet rim to characterize the α/β-cell distribution irregularity. The larger the index value, the greater the deterioration of the islet structure.

## Results

We explain the integration analysis of the rat and human data sets step by step as follows.

### Diabetic causal factor induces a clear contrast in the sample loadings of the principal eigen-component

After the raw RNA-seq and microarray data were normalized, we performed SVD and dual eigen-analysis of the two transcriptomic profiles of rat islets and human islets. SVD stratified each expression profile into orthogonal components by their singular values from high to low.

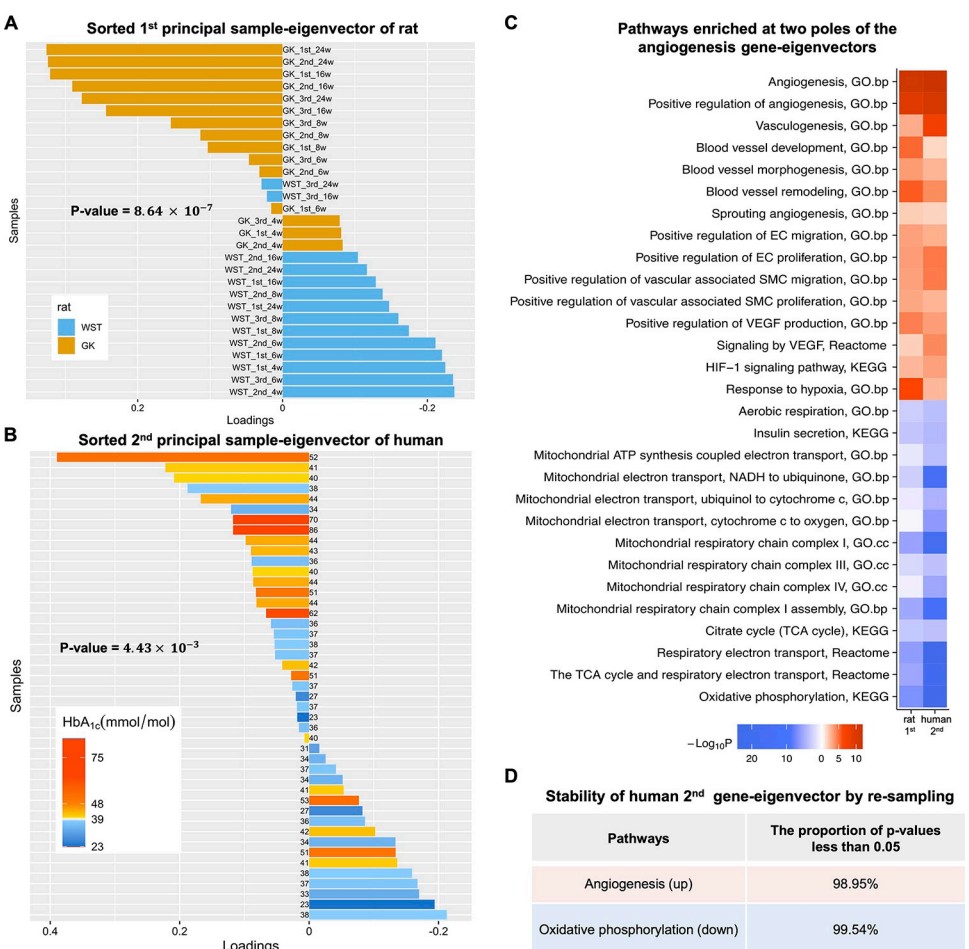

**Fig 2. The common principal gene-eigenvector characterized by two poles of angiogenesis versus oxidative phosphorylation corresponds to the sample loadings with a contrast respectively of the genetic factor in rat, and of certain type 2 diabetic pathological indices in human.** (**A**) Sorted loadings of the first principal sample-eigenvector of rat. The genetic factors are indicated by the bar color. GK samples are mostly at the top half whereas WST samples are mostly at the bottom half. The p-value of such a contrast is $8.64 \times 10^{-7}$. (**B**) Sorted loadings of the second principal sample-eigenvector of human. The p-value of the contrast between samples with HbA$_{1c}$ levels higher and lower than 39 mmol/mol (5.7%) is $4.43 \times 10^{-3}$. The HbA$_{1c}$ levels (the digit next to each bar) are indicated by the bar color depth. (**C**) Heatmap showing pathways enriched at two poles of the angiogenesis gene-eigenvectors. Rat, the first principal gene-eigenvectors; human, the second principal gene-eigenvector; red, positive pole; blue, negative pole. EC, endothelial cell; SMC, smooth muscle cell; VEGF, vascular endothelial growth factor. (**D**) Stability of the enriched pathways in the second principal gene-eigenvector of human by re-sampling islets.

S1 Table shows the top three singular values of the two profiles and their proportions in the total sums, which measure their relative contributions to the total mRNAs. The top eigenvectors of SVDs were expected to capture the principal information in the expression profiles. The top two components explain 38.50% of the sum of squares in rat while they do 31.16% in human.

Hereafter, we refer to the samples of GK rats as T2D susceptible samples, abbreviated to DS samples. We refer to islets from human donors with abnormal HbA$_{1c}$ levels ($\geq$39 mmol/mol or $\geq$5.7%) and with normal values respectively as AH and NH samples. The GK genetic factor induced a statistically significant contrast between DS and control samples in the sample loadings of the first principal eigen-component (Fig 2A). A perfect contrast could be obtained by a more careful selection of samples for SVD (S4 Fig). Although the human islets data were from

an observational study, the contrast between AH and NH samples in terms of their loadings of the second principal eigen-component was statistically significant (Fig 2B). For the sake of clarity, we set the pole of DS samples in the rat's polarized eigenvector, and AH samples in the human's eigenvector to be the positive one.

In this report, significant contrasts between DS and control samples of rat, and between AH and NH human samples were indeed observed in one principal eigen-component. It is interesting to know if their dual gene-eigenvectors share any transcriptomic features in pancreatic islets.

## Angiogenesis is both a prominent and conserved feature in the principal gene-eigenvector, in rat and human

An apparent feature shared by the two species is that the angiogenesis and oxidative phosphorylation pathways were enriched respectively at the positive and negative poles of the dual gene-eigenvectors (Fig 2C), indicating that the angiogenesis was up-regulated in the type 2 diabetic islets while oxidative phosphorylation was down-regulated.

The GO biological process "angiogenesis" and its related processes, such as "vasculogenesis", "sprouting angiogenesis", "blood vessel development", "blood vessel morphogenesis", and "blood vessel remodeling", were all up-enriched in rat's first and human's second principal gene-eigenvectors (Fig 2C), which we refer to as the angiogenesis gene-eigenvectors hereafter. More specifically, angiogenesis-related processes were enriched at the positive pole corresponding to DS samples in rat, and to AH samples in human.

The angiogenesis process is composed of two phases [32]. The first phase is activation, when endothelial cells (ECs) are activated to undergo migration, branching, proliferation, and lumen formation. The second phase is resolution, during which EC proliferation terminates while mural cells, called pericytes in capillaries or vascular smooth muscle cells (SMCs) in larger vessels, are induced to differentiate and are recruited to the newly formed vessels to establish the vessel wall. The GO biological processes involved in both phases, such as "positive regulation of EC migration", "positive regulation of EC proliferation", "positive regulation of vascular associated SMC migration", and "positive regulation of vascular associated SMC proliferation", were indeed up-enriched in the angiogenesis gene-eigenvectors of the two species (Fig 2C).

## Consistency of the RNA-seq/microarray expression with RT-PCR results

The RNA-seq data of GK/WST rats were validated by the qRT-PCR method [9]. The expression of 6 randomly selected genes was measured by qRT-PCR method at weeks 4, 8 and 16, as were shown in S6(C) Fig in [9]. The RNA-seq time course data of these 6 genes by averaging over the three replicates at each time point were shown in S5 Fig. Moreover, the fold changes of some genes comparing 4-month-old GK and Wistar rats by RT-PCR were reported [33]. Among them, we selected 4 genes involved in angiogenesis, namely, *TIMP1*, *FN1*, *DCN*, *MMP14*. Then we compare the RNA-seq data of the 16-week-old rats with the RT-PCR data in S6 Fig. The results were quite consistent.

The expression of several genes in the human islet samples was validated by the qRT-PCR method as well [34].

## Oxidative phosphorylation stands at the opposite pole of the angiogenesis gene-eigenvector

The KEGG pathway "oxidative phosphorylation", the REACTOME pathway "tricarboxylic acid (TCA) cycle and respiratory electron transport", and the GO cellular components

"mitochondrial respiratory chain complex I and III" were all enriched at the opposite poles of the angiogenesis gene-eigenvectors from the two species (Fig 2C). It indicated that the oxidative phosphorylation activities were reduced in the type 2 diabetic islets.

The process of oxidative phosphorylation in mitochondria is coupled with the synthesis of ATP. Mitochondria can control the secretion of insulin in an ATP-dependent manner, not only by providing energy in the form of ATP, but also by triggering closure of the ATP-sensitive potassium channel ($K_{ATP}$). The closure of $K_{ATP}$ channel results in depolarization of the plasma membrane, which triggers the opening of the voltage-gated $Ca^{2+}$ channel. This results in an influx of $Ca^{2+}$ into the cell that drives the exocytosis of insulin granules [35]. In concert with the reduced oxidative phosphorylation, the KEGG pathway "insulin secretion" was down-enriched as observed in the angiogenesis gene-eigenvectors (Fig 2C).

### Stability of the principal gene-eigenvector

Unlike the GK/WST islets, the human islets data are from an observational study. We propose to evaluate the stability of its principal gene-eigenvectors by re-sampling the islets. Namely, we compare the SVD of all islets with that of random sample of size 56 (90% of the original size), and repeat for 10000 times. The correlations of the top two principal gene-eigenvectors are displayed by the density plots in S7 Fig. According to the enrichment analysis with a p-value of 0.05 as the significance threshold, angiogenesis was up-regulated in 99.0% samples while oxidative phosphorylation was down-regulated in 99.6% cases (Fig 2D).

### Similar principal eigen-component observed in the unified SVD of the concatenated expression profile

To integrate the expression profiles of rat and human directly in the SVD step, we concatenated the two expression profile matrices by aligning their homologous genes (Fig 3A). Specifically, we identified a total of 11,146 homologous genes. After equalizing their variations, we concatenated them into a single matrix. Next, we applied the SVD and dual eigen-analysis procedure to the concatenated expression matrix. In the first sample-eigenvector of the concatenated expression matrix, the positive and negative loadings of samples from both rats and human correspond to T2D and normal samples roughly and respectively. Furthermore, we observed significant contrasts in both the rat and human components in the sample loadings (Fig 3B and 3C) similar to those obtained from the species-alone SVDs (Fig 2A and 2B). The enrichment result of the positive pole of the first gene-eigenvector showed the significantly up-regulated GO biological process "angiogenesis", along with other related pathways (Fig 3D). The results are similar to those obtained from the species-alone SVDs (Fig 2C). In the opposite pole, we observed the significantly down-regulated KEGG pathway "oxidative phosphorylation" (Fig 3D), as well as other associated pathways shown in Fig 2C. The similar results were obtained by applying the unified SVD to the concatenated expression matrix without variation equalization.

### Progression of transcriptional angiogenesis and oxidative phosphorylation over time in GK rat islets

Since the T2D development in GK rats are fairly homogeneous, the results of dual eigen-analysis can be checked by directly comparing the expression of GK and WST rats at each week. Indeed, Fig 4A shows an up-trend of angiogenesis over time, which is in line with the sample loadings (Fig 4B) that couple with the rat's angiogenesis gene-eigenvector. Fig 4A also shows the substantial reduced oxidative phosphorylation after week 8. It is consistent with the

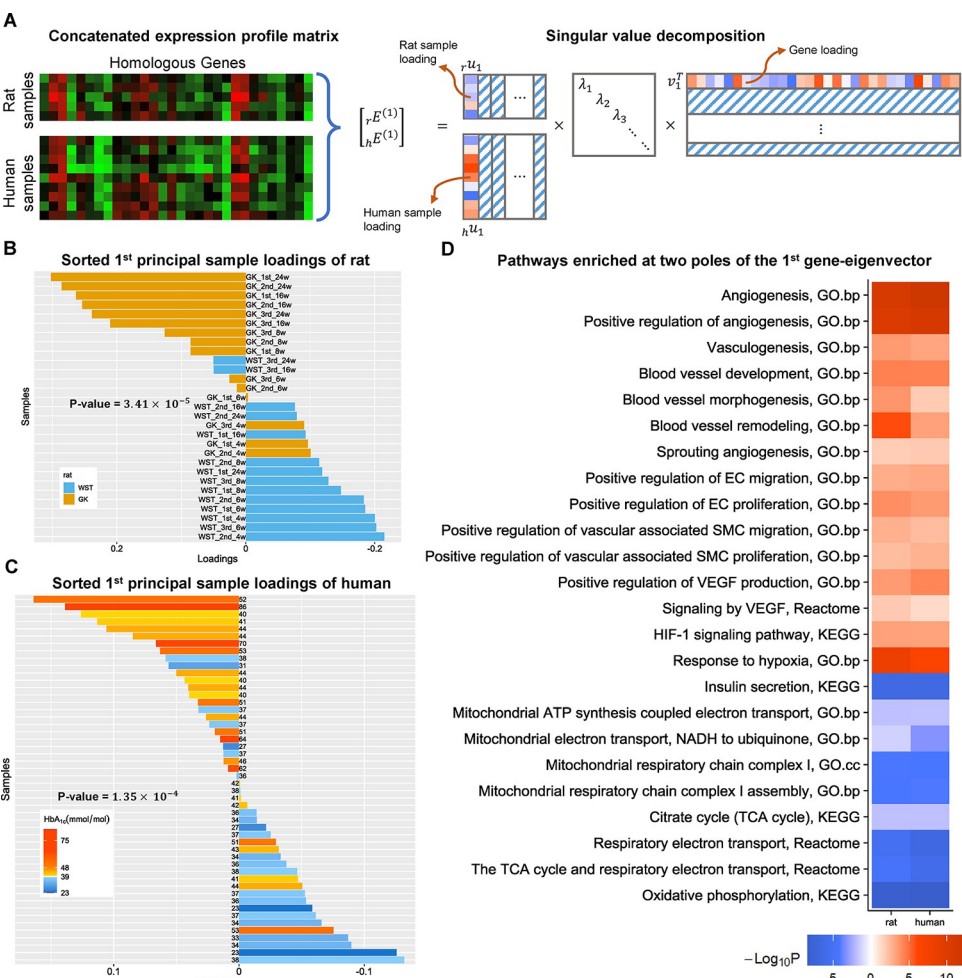

**Fig 3. SVD and dual eigen-analysis of the concatenated expression profile of rat and human. (A)** Illustration of the SVD procedure. After removal of baseline and equalization of variations, the expression matrices of rat and human denoted respectively by $_rE^{(1)}$ and $_hE^{(1)}$ are concatenated into one single matrix by aligning their homologous genes. SVD was carried out on the concatenated expression profile matrix. **(B)** The sorted loadings of rat in the first principal sample-eigenvector. The genetic factors are indicated by the bar color. GK samples are mostly at the top half whereas WST samples are mostly at the bottom half. The p-value of such a contrast is $3.41 \times 10^{-5}$. **(C)** The sorted loadings of human in the first principal sample-eigenvector. The p-value of the contrast between samples with HbA$_{1c}$ levels higher and lower than 39 mmol/mol (5.7%) is $1.35 \times 10^{-4}$. The HbA$_{1c}$ levels (the digit next to each bar) are indicated by the bar color depth. **(D)** Heatmap: Significances of the pathways enriched at two poles of the first gene-eigenvector. The pathway enrichment of rats and humans were respectively based on their own gene sets; red, positive pole; blue, negative pole. EC, endothelial cell; SMC, smooth muscle cell; VEGF, vascular endothelial growth factor.

composite result of rat's first and second gene-eigenvector that represents the compensatory mechanism during the rat development state, as indicated by its dual sample-eigenvector, see subsection "Compensatory insulin generation during the early stage in GK islets".

## Alteration of α/β-cell distribution over time in GK islets

Fig 4C shows the islet micrographs from 6-month-old Wild-type (WT) at top and Akt1$^{+/-}$Akt2$^{-/-}$ male mice at bottom. The distribution of the distances of glucagon pixels away from the islet rim is shown on the right. The kernel density estimation (KDE) of the distances roughly represented the spatial distribution of α- and β-cells.

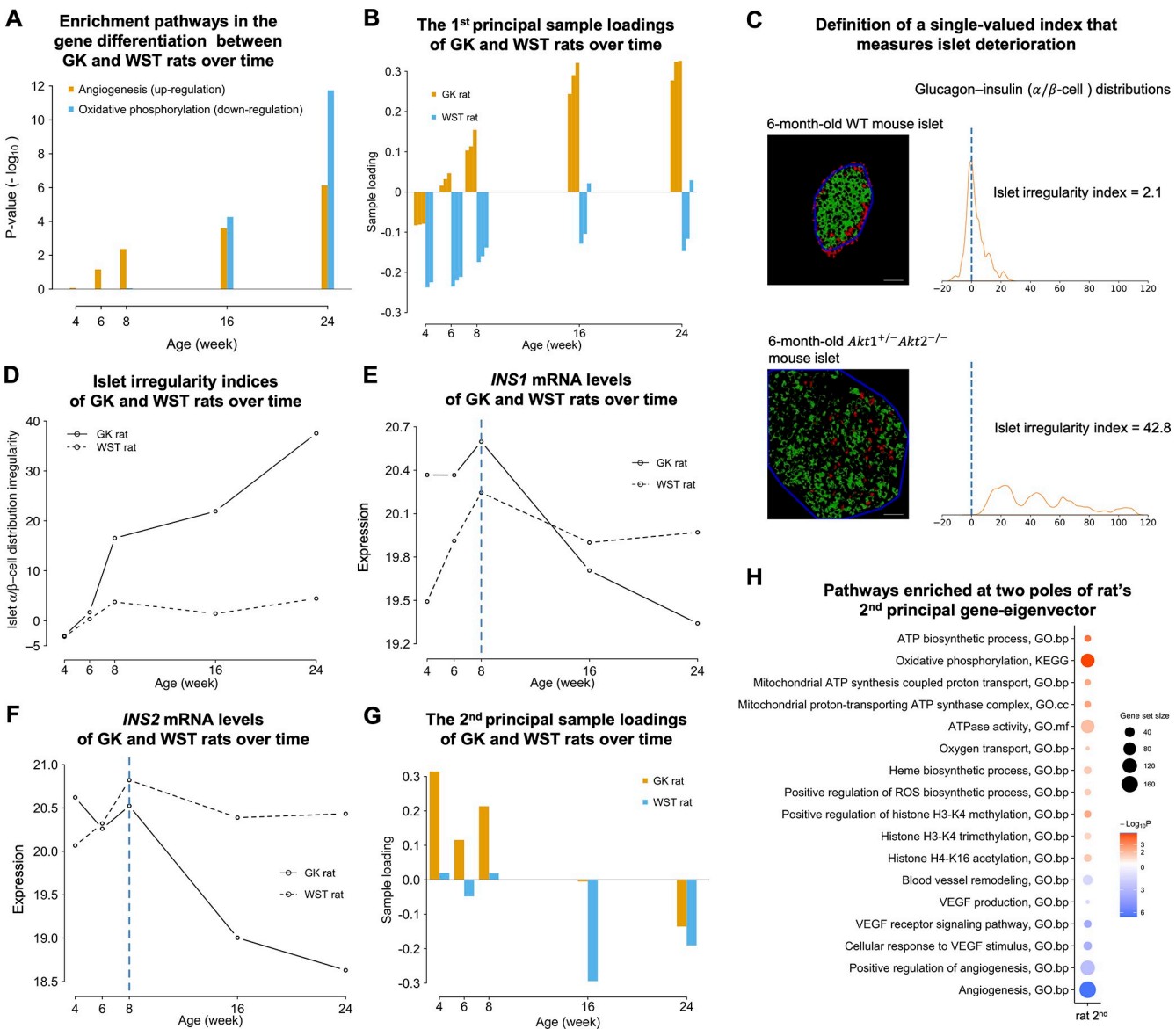

**Fig 4. The insulin (mRNA), morphological, and molecular changes along the T2D progression of GK rats.** (A) Enrichment of the angiogenesis pathway in the gene differentiation between GK and WST rats over the five time points. (B) The first principal sample loadings of GK and WST rats as displayed by each time point. (C) Definition of a single-valued morphometric index that measures islet deterioration, as exemplified by the islet micrographs of 6-month-old wild-type (WT) and Akt1[+/−]Akt2[−/−] (a type 2 diabetic model) mice. In each islet micrograph (scale bar shown at the bottom is 50 μm), the rim (blue line) of the insulin region (green) is sketched, and the distances of glucagon pixels (red) away from the rim are defined. Pixels inside the rim have positive distances and those outside have negative ones. A kernel density of the distances is shown next to each micrograph. It quantifies the spatial distribution of α- and β-cells via that of glucagon and insulin. The islet irregularity index is defined by the trimmed mean of the kernel density. (D) Islet α/β-cell distribution irregularity over time in GK and WST rats. (E) *INS1* mRNA levels of GK and WST rats. At each time point is the average value of 3 individuals. (F) *INS2* mRNA levels of GK and WST rats. (G) The second principal sample loadings of GK and WST rats. (H) Heatmap shows pathways enriched at the two poles of the second principal gene-eigenvector of rat. ATP biosynthetic process, oxygen transport, and oxidative stress-related pathways are enriched at the positive pole (red), and angiogenesis-related pathways at the negative pole (blue). The increased insulin levels in GK rats during week 4–8 correspond to up-regulation of ATP synthesis and down-regulation of angiogenesis. ROS, reactive oxygen species; VEGF, vascular endothelial growth factor.

The same distributions were obtained for GK and WST rats in S8 Fig. The KDE curves obtained from GK islets in the early stage (4–6 weeks) were similar to those from WST islets, nearly distributed around zero, indicating α-cells mostly distributed at the islet rim in normal islets. In comparison, the mass of KDE curves obtained from GK islets in the late stage (8–24

weeks) progressively shifted towards the positive direction and became flatter, indicating that α-cells penetrated into islets during the T2D progressions.

## Islet irregularity index and its synchronous alterations with transcriptional angiogenesis

In the example in Fig 4C, the islet irregularity index of the wild-type mouse was 2.1 whereas that of the diabetic mouse was 42.8.

The indices remained below 5 for WST islets and gradually increased up to 37.5 in GK islets (Fig 4D). The Wilcoxon signed-rank test for comparing the indices of GK and WST islets resulted in a p-value of 0.031. The differences between the indices of GK and WST islets at the five time points are increasing, demonstrating that the islet irregularity worsens along the T2D progression.

The alteration of the islet irregularity indices in GK islets is in synchrony with that of transcriptional angiogenesis (Fig 4A). From the perspective of dual eigen-analysis, the Spearman's and Pearson's correlation coefficients between the islet irregularity indices and the sample loadings coupled with the rat's angiogenesis gene-eigenvector (Fig 4B) were equal to 1 and 0.95 respectively. The synchronous changes allow us to predict the α/β-cell distribution irregularity at the morphological level by the progression of transcriptional angiogenesis at the molecular level, and *vice versa*.

## Synchronous deterioration of oxidative phosphorylation and insulin levels

To further investigate the pathological implications of transcriptional angiogenesis, we considered its association with the time-course data of insulin mRNA levels at five time points (weeks 4, 6, 8, 16, and 24) from the same data. *INS1* and *INS2* mRNA levels of GK and WST rats are respectively shown in Fig 4E and 4F. The value at each time point is the average of 3 individuals. *INS1* shows an up-trend before week 8 and a sharp down-trend after week 8 in GK islets; while in WST islets, after a significant up-regulation before week 8, the expression moderately drops down and stays stable in weeks 16 and 24. *INS2* shows a similar sharp down-trend after week 8 in GK islets and a relatively stable pattern in WST islets.

It is noted that plasma insulin levels were measured for GK and Wistar-Kyoto (WKY, a substrain of Wistar) rats [36] at ages: 4, 8, 12, 16, and 20 weeks. The time-course averages of six rats at each time point were shown in S9 Fig. The pattern of an up-trend before week 8 and a down-trend after week 8 observed in GK rats is similar to that of insulin mRNA levels in Fig 4E.

The reduction of insulin levels turns out to be synchronous to that of oxidative phosphorylation at the transcriptional level (Figs 4A, 4E, 4F and S9). Oxidative phosphorylation in mitochondria is coupled with the synthesis of ATP, which plays a key role in insulin generation and secretion. From the perspective of dual eigen-analysis, the sample loadings of the first eigenvector of GK rats go up significantly after week 8 (Fig 4B), with the down-regulation of oxidative phosphorylation in its dual gene-eigenvector (Fig 2C). Thus, the first eigenvectors can explain the down-trend of insulin levels after week 8 to some extent. The heightened level before week 8 can be explained by the second eigenvector.

## Compensatory insulin generation during the early stage in GK islets

When we examined rat's second principal eigen-component, a contrast before week 8 and after week 16 could be observed in the sample loadings of GK rats (Fig 4G). The sample loadings were positive from week 4 to week 8, during which the insulin level gradually increased.

Then the insulin mRNA level dropped down (Fig 4E and 4F), corresponding to the negative sample loadings after week 16 (Fig 4G).

When we checked the second gene-eigenvector, the angiogenesis-related pathways were down-enriched while the ATP biosynthetic process was up-enriched for the samples up to week 8 (Fig 4H). This timing coincided with the increase of insulin level up to week 8 (Fig 4E and 4F). Moreover, the up-regulated GO biological processes "oxygen transport" and "heme biosynthetic process" (Fig 4H) indicated that oxygen needed for insulin secretion was compensated during this period. This is in line with the report that increased islet blood flow is present in young diabetic GK rats [37]. In addition, the GO biological process "positive regulation of reactive oxygen species (ROS) biosynthetic process" was up-regulated. Accordingly, we observed up-regulated histone modifications such as H3-K4 methylation and H4-K16 acetylation, as epigenetic responses to oxidative stress [38,39]. Notably, the KEGG "HIF-1 signaling pathway" was not active ($p = 0.69$).

## Gene-eigenvector aggregated from two species

To further investigate the angiogenesis gene-eigenvector shared by the two species, we used the Robust Rank Aggregation method [30] to integrate the two eigenvectors into one gene list, in which those homologous genes were ranked by their significances. It is referred to as aggregated gene-eigenvector hereafter. We performed enrichment analysis on the aggregated gene-eigenvector. Some of the pathway enrichment results are shown in Fig 5A.

In the aggregated gene-eigenvector, we found 130 significantly up-regulated genes ($p<0.05$) were involved in the angiogenesis-related GO biological processes (S2 Table). Two major types of angiogenic molecules are growth factors and matrix metalloproteinases (MMPs). The isoform and receptor genes of multiple angiogenic growth factors, including VEGF, FGF2, PDGF, HGF, CTGF and TGF-β, ranked significantly high in the aggregated gene-eigenvector. Multiple MMP family member genes were significantly up-regulated, such as *MMP1* (rank 117 in human, $p = 0.021$), *MMP2* (rank 91, $p = 1.70\times10^{-3}$), *MMP3* (rank 21 in human, $p = 3.75\times10^{-3}$), *MMP7* (rank 259, $p = 6.84\times10^{-3}$), *MMP10* (rank 11 in human, $p = 1.96\times10^{-3}$), and *MMP14* (rank 321, $p = 8.74\times10^{-3}$). Moreover, a collection of angiogenic EC marker genes were significantly up-regulated (S3 Table), such as *CD44* (rank 27, $p = 2.61\times10^{-4}$), *VCAM1* (rank 4, $p = 1.53\times10^{-5}$), *CD93* (rank 128, $p = 2.84\times10^{-3}$), *KLF4* (rank 268, $p = 7.00\times10^{-3}$), *TIE2* (rank 1119, $p = 0.040$), and *ADAM9* (rank 352, $p = 9.85\times10^{-3}$). These genes together with their aggregated ranks are summarized in Fig 5B. The detail of the above genes as well as other angiogenesis-relating genes can be found in S2 Text.

## Transcriptional angiogenesis and HIF signaling pathway

The hypoxia-inducible factor (HIF) pathway is currently viewed as a master regulator of angiogenesis, and most transcriptional responses to low oxygen are mediated by HIFs [40]. Several significantly up-regulated angiogenesis genes in the aggregated gene-eigenvector were identified as downstream of the KEGG "HIF-1 signaling pathway", such as *TIE2* (rank 1119, $p = 0.040$), and *PAI1* (rank 120, $p = 2.56\times10^{-3}$). Multiple genes in the VEGF signaling pathway were up-regulated as well (S4 and S5 Tables). With a close look at the KEGG "HIF-1 signaling pathway", which was up-enriched in the angiogenesis gene-eigenvectors of both species (Fig 2C), we found 23 genes were significantly up-regulated in the aggregated gene-eigenvector (S6 Table).

Notably, the expression of *HIF-1α* itself was significant as well (rank 657, $p = 0.021$), and so is *HIF-2α*, also known as *EPAS1* (rank 829, $p = 0.029$). Under hypoxia, HIF-1α or HIF-2α

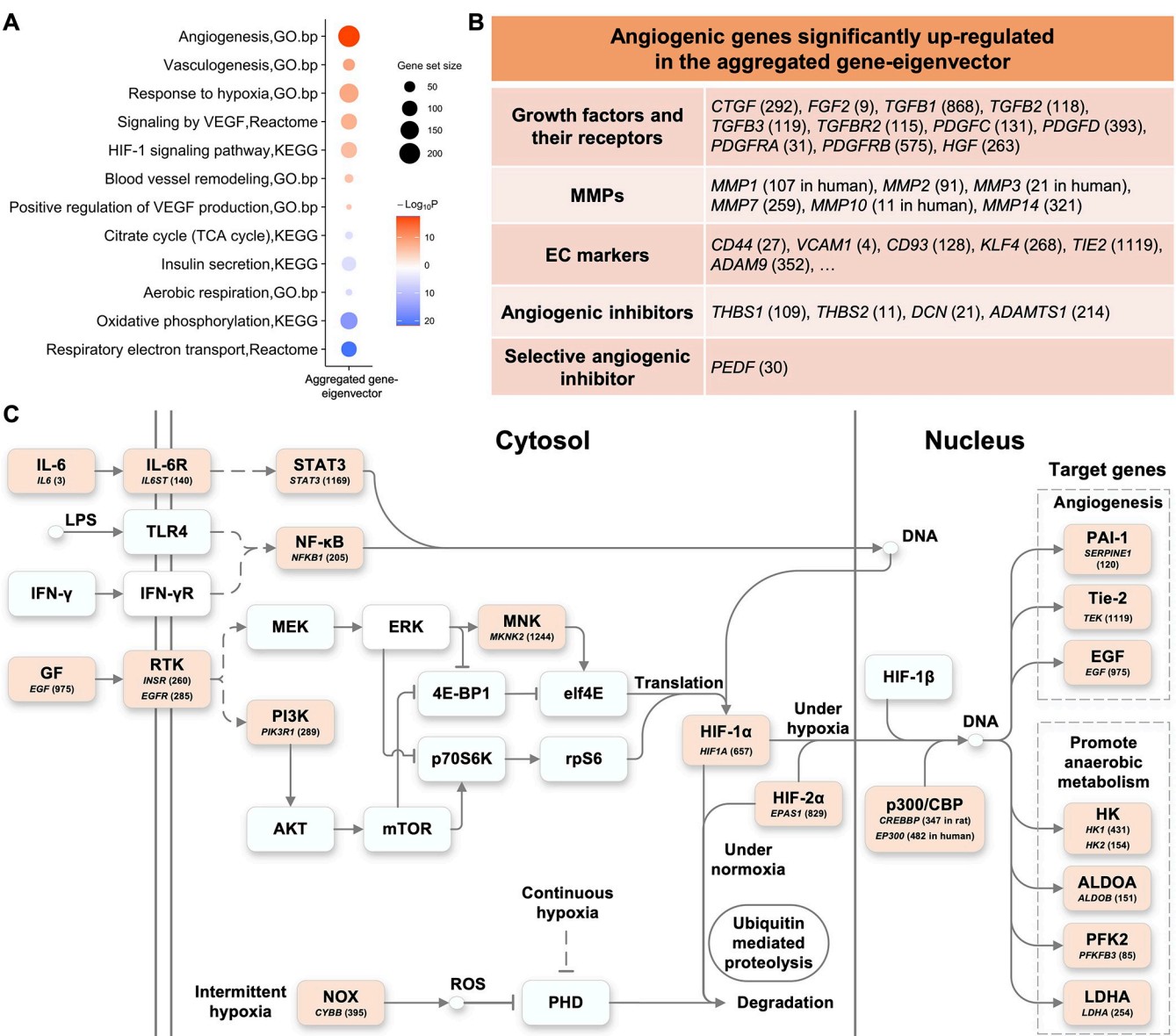

**Fig 5. The angiogenesis gene-eigenvectors of rat and human are integrated by the Robust Rank Aggregation method into the aggregated gene-eigenvector, in which those homologous genes are ranked by their significances.** (A) The enrichment biological pathways in the aggregated gene-eigenvector. (B) The angiogenesis related genes significantly up-regulated in the aggregated gene-eigenvector. The genes encode growth factors and their receptors, MMP family members, EC markers, and angiogenic inhibitors. Their aggregated ranks are shown in parentheses. (C) Superimposition of up-regulated mRNA levels on the HIF signaling pathway. The ranks of the high-ranking genes in the HIF signaling pathways are superimposed to the nodes they belong to. These nodes are marked in color. These genes include the cytokine *IL6* ($p = 7.72 \times 10^{-6}$), the growth factor *EGF* ($p = 0.035$); the receptors *INSR* ($p = 6.84 \times 10^{-3}$), *EGFR* ($p = 7.52 \times 10^{-3}$); the signal transducer and transcriptional activator downstream of IL6, *STAT3* ($p = 0.042$); the signaling molecules *PIK3R1* ($p = 7.67 \times 10^{-3}$), *MKNK2* ($p = 0.045$), *CYBB* ($p = 0.011$) that generates superoxide. HIF-2α (*EPAS1*, $p = 0.029$) is included along with HIF-1α ($p = 0.021$), which would be degraded under normoxia. Under hypoxia, HIF-1α or HIF-2α recruits coactivators such as CREBBP and EP300, heterodimerizes with ARNT (HIF-1β), and regulates numerous hypoxia-inducible target genes. The key target genes involved in angiogenesis and in promoting anaerobic metabolism are shown on right. The aggregated ranks of relevant genes are shown in parentheses.

heterodimerizes with ARNT (HIF-1β), and regulates numerous hypoxia-inducible genes [41]. The key genes are summarized in Fig 5C and S6 Table.

It is noted that a group of up-regulated genes in the aggregated gene-eigenvector, also as downstream of the KEGG "HIF-1 signaling pathway", were involved in promoting anaerobic

metabolism (Fig 5C). They included *HK1* (rank 431, $p$ = 0.012), *HK2* (rank 154, $p$ = 3.45×10$^{-3}$), *ALDOB* (rank 151, $p$ = 3.39×10$^{-3}$), *PFKFB3* (rank 85, $p$ = 1.52×10$^{-3}$), and *LDHA* (rank 254, $p$ = 6.67×10$^{-3}$). We observed a metabolism switch in the gene signature, as shown by the aggregated gene-eigenvector, from mitochondrial aerobic respiration to activities in anaerobic conditions, or fermentation (S3 Text).

In the study [42], *HIF-1α* was knocked down in Min6 cells by RNAi, and led to the significant decreased expression of *ALDOB* and *PFK*, involved in the anaerobic metabolism. Similar results were reported for β-HIF-1α-null mice. Both β-HIF-1α-null mice and HIF-1α knockdown Min6 had markedly impaired first-phase glucose-stimulated insulin secretion.

Apart from the 23 up-regulated genes involved in the KEGG "HIF-1 signaling pathway", 24 significantly up-regulated genes were related to hypoxia. S7 Table shows the aggregated ranks of these hypoxia response genes. Moreover, the GO biological process "aerobic respiration" was down-enriched in the angiogenesis gene-eigenvectors of the two species while the GO biological process "response to hypoxia" was up-enriched (Fig 2C). Notably, Fig 3 in the study [37] showed islet blood flow for three types of rats at three time points, 5, 12, and 52 weeks. GK rats demonstrated increasing blood flow between 5 and 12 weeks and followed by a pronounced decrease after 12 weeks. Meanwhile, the transcriptional angiogenesis was not significant before week 6 and became very pronounced after week 16 (Fig 4A). In contrast, the islet blood flow keeps at augmented levels in older WST rats. Taken together, during the progression of T2D, abnormal islet angiogenesis, at the transcriptional level, is likely to be induced directly by hypoxic stress.

## Inflammation and angiogenesis

It has been reported that inflammation has a reciprocal interaction with angiogenesis [43], and islet inflammation plays an important role in the pathogenesis of β-cell failure in T2D [44]. Thus, we examined the inflammation-associated pathways including the NOD-like receptor, tumor necrosis factor (TNF), nuclear factor-kappa B (NF-κB) signaling pathways, as well as the immune response-related pathways. Indeed, plenty of them were significantly up-enriched at the angiogenesis gene-eigenvectors of the two species (S8 and S9 Tables). In particular, *NFKB1* (rank 205, $p$ = 5.11×10$^{-3}$) is also a member in the KEGG "HIF-1 signaling pathway".

## Anti-angiogenic genes and protective mechanism

If the angiogenesis and alteration of α/β-cell distribution in islets are adverse to the long-term benefit, then a certain preventive mechanism is expected to exist. Indeed, we identified several anti-angiogenic genes significantly up-regulated in the aggregated gene-eigenvector, including *THBS1* (rank 109, $p$ = 2.28×10$^{-3}$) and *THBS2* (rank 11, $p$ = 1.05×10$^{-4}$). A major pathway by which THBS1 or THBS2 inhibits angiogenesis involves an interaction with CD36 on ECs, which leads to apoptosis of both the liganded and adjacent cells [45]. THBS1–deficient mice were glucose intolerant, showed a reduction in glucose-stimulated insulin release, despite having an increased β-cell mass [46].

In addition, two anti-angiogenic genes, *DCN* and *SERPINF1*, respectively ranked 21 and 30 in the aggregated gene-eigenvector. Their functions are helpful for understanding the phenomenon of thickened and fragmented capillaries. *SERPINF1* encodes the protein pigment epithelium-derived factor (PEDF), a potent and endogenous angiogenesis inhibitor [47]. Its anti-angiogenic activity is selective in that PEDF destroys newly forming vessels but does not appear to harm the existing one [48]. *DCN* encodes the protein decorin, which can exhibit either a pro-angiogenic or an anti-angiogenic activity; nevertheless, in pathological conditions such as tumorigenesis and various inflammatory processes, the role of decorin in angiogenesis

is mainly inhibitory [49]. Decorin and PEDF both have protective effects against diabetes complications in multiple organs and tissues, including retinopathy, nephropathy, and cardiac diseases (S4 Text).

The high expression of the angiogenesis inhibitors, together with their protective roles, on the one hand, confirm the pressure of islet angiogenesis along the T2D progression, and on the other hand, imply that a counteractive mechanism does exist in diabetic islets.

## The first and second (angiogenesis) sample-eigenvectors jointly account for most human cases with high HbA$_{1c}$ levels

Other than the common angiogenesis-oxidative phosphorylation gene-eigenvector, which was the second one of the human expression data, we further examined the first principal gene-eigenvector. At one of the two poles, we observed the down-enriched "insulin processing" and "insulin secretion" pathways that implied β-cell dysfunction, and the down-enriched "insulin signaling pathway" that implied insulin resistance. At the same pole, the pathways inducing cell death, such as "apoptosis", "necroptosis", and "TP53 regulates transcription of cell death genes", were up-enriched (Fig 6A), and they might be related to the mass loss of β-cells in T2D. These bioinformatics results suggested that the first principal gene-eigenvector of human represented a rather late stage of T2D.

On the sample loading side, we can predict the samples' HbA$_{1c}$ levels by the top two eigenvectors. Shown in Fig 6B is the bubble plot of human samples on the plane with x- and y-coordinates corresponding respectively to the loadings of the first and second principal sample-eigenvectors. Next, we divided the plane into four regions by the x-value -0.01 and y-value 0.025 respectively along the horizontal and vertical direction. 92.31% AH samples fell into region I, II, or III; namely, either their first loadings were greater than -0.01 or their second loadings were greater than 0.025. Moreover, 80% samples in region I were AH samples whereas only 18% in region IV were so. The average HbA$_{1c}$ level of samples in region I was 48 mmol/mol (6.5%) whereas that in region IV was 34 mmol/mol (5.3%).

The first principal sample-eigenvector alone did have some ability in discriminating AH samples from NH ones. That is, the loadings of AH samples were overall larger than those of

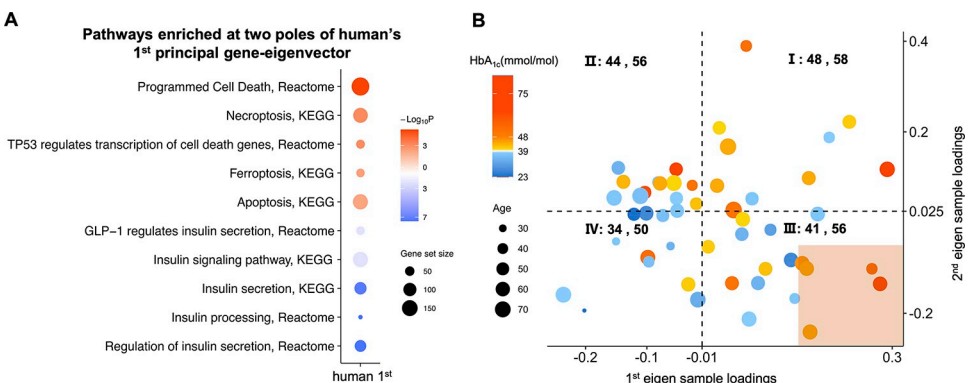

**Fig 6. Human's principal eigenvectors. (A)** Heatmap showing pathways enriched at two poles of the first principal gene-eigenvector of human. **(B)** The bubble plot of human samples on the plane with x- and y-coordinates corresponding respectively to loadings of the first and second principal sample-eigenvectors. The plane is divided into four regions, I, II, III, and IV. The average HbA$_{1c}$ and age levels of samples in each region are shown next to the region label. The high HbA$_{1c}$ levels of samples in the shaded area can be explained by the first principal eigen-component, which reflects the impaired insulin-related functions (Fig 6A). The remaining samples are alternatively displayed in Fig 2B in the descending order of the second principal sample loadings. The HbA$_{1c}$ levels are indicated by the dot colors whereas the age levels are indicated by the dot sizes.

NH samples in the sense that the p-value of the Wilcoxon rank-sum test was 0.017 (S10 Fig), and the Spearman's correlation coefficient between the first principal sample loadings and the HbA$_{1c}$ levels is 0.33 with a p-value of $9\times10^{-3}$. Marked in the shaded area in the bubble plot of Fig 6B are five AH samples, whose first sample loadings were large yet second loadings were small. In other words, their high HbA$_{1c}$ levels were explained by the first principal gene-eigenvector but not by the second, or the angiogenesis one. The enrichment results implied that these five samples no longer had angiogenesis (S10 Table). After they were excluded, the angiogenesis sample-eigenvector could discriminate between AH and NH samples with a higher significance. That is, the comparison of their second principal sample loadings by Wilcoxon rank-sum test resulted in a p-value of $4.43\times10^{-3}$ (Fig 2B), and the Spearman's correlation coefficient between the second principal sample loadings and the HbA$_{1c}$ levels is 0.41 with a p-value of $2.4\times10^{-3}$.

### Recapitulation of the data integration approach

As shown on the left in Fig 7A, the expression and quantified micrograph data from the designed experiment can be analyzed by the two-sample enrichment tests over time. Any statistically significant differences such as angiogenesis related gene expression and α/β-cell distribution are due to the genetic causal factor as in the treatment-control studies (Fig 4A and 4D). However, the SVD of the whole transcriptome offers a more profound systems biology result. That is, the up/down-regulated angiogenesis/oxidative phosphorylation are among the most prominent biological processes enriched in the principal gene-eigenvector. The corresponding dual sample-eigenvector shown on the right top of Fig 7A not only confirmed the contrast induced by the genetic causal factor, but also offer a longitudinal spectrum of the progression (Fig 2A).

The two datasets were integrated through the aggregated gene-eigenvector shown in the middle of Fig 7A. Its stability on the human islets side was demonstrated by the re-sampling results. The results indicate rat and human share certain molecular mechanism in the T2D development. The dual sample-eigenvector shown on the right bottom of Fig 7A demonstrated the association between the molecular alterations in the gene-eigenvector with the heightened HbA$_{1c}$ levels in human.

Notably, the aggregated gene-eigenvector corresponds respectively to the first principal component of the GK/WST islets and the second principal component of the human islets. The data of the GK/WST rats were collected up to week 24, namely, during the relatively young age in their lifespans. It is not unexpected that the second principal component reflects the compensatory mechanism that resists the T2D during the development stage (Figs 4H and 7B). In contrast, the median age of the human donors is 57 years old. Insulin secretion was severely impaired in the first principal gene-eigenvector, indicating that it corresponds to relatively late stage of diabetes (Figs 6A and 7B). The stratification of the samples by SVD identified both the common features and the differences in the two data sets (Fig 7B).

### The integrative analysis unravels the principal transcriptional alterations underlying the islet deterioration of morphology and insulin secretion along T2D progression

In the case of GK/WST islets, at the macro-level we have quantified the alteration of islet morphology and examined the insulin levels. We highlight their synchronous and principal alterations at the transcriptional micro-level in Fig 7C.

Shown at the top panel of Fig 7C are the alterations of α/β-cell distribution and abnormal transcriptional angiogenesis. The morphological alterations in terms of α/β-cell distribution

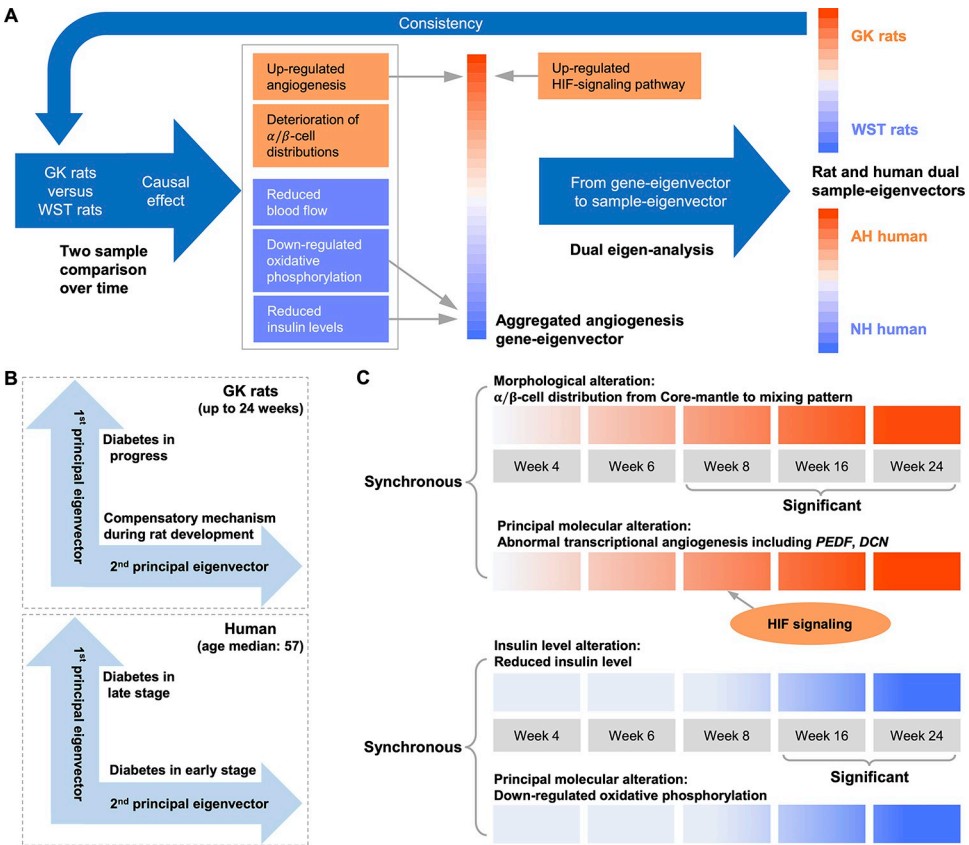

**Fig 7. Recapitulation of the data integration approach. (A)** The left side includes the results of two-sample methods that used to analyze the designed experiment data of GK/WST rats. These results are consistent with the aggregated gene-eigenvector in the middle. By the dual eigen-analysis, the gene-eigenvector corresponds to the sample-eigenvectors that are on the right side. The contrast induced by the sample-eigenvectors is consistent with the genetic factors of GK/WST rats. **(B)** The first principal component of GK/WST islets reflects the process of T2D, while the second one shows the compensatory mechanism that resists the T2D during the development stage. The first principal gene-eigenvector of human islets corresponds to relatively late stage of diabetes, while the second one corresponds to early stage. Notably, the first principal component of rats and the second principal component of human is identified as the shared signature of T2D progression. **(C)** The synchronous alterations of the GK rats in the T2D progression. The top panel includes alterations of the α/β-cell distribution at the morphological level and of the abnormal transcriptional angiogenesis at the molecular level. These alterations are significant from week 8. The bottom panel includes alterations of the insulin levels and the transcriptional oxidative phosphorylation at the molecular level. These alterations are significant from week 16.

are from core-mantle to a mixing pattern. The abnormal transcriptional angiogenesis includes *THBS1*, *THBS2*, *DCN*, *PEDF*, which are most prominent in the principal molecular component. The alterations in GK islets with respect to WST islets became significant from week 8 (Fig 4A and 4D).

Shown at the bottom panel of Fig 7C are the alterations of insulin levels and transcriptional oxidative phosphorylation. It is demonstrated that the mRNA insulin levels from this study were consistent with blood insulin levels reported in the literature (Figs 4E, 4F and S9). Meanwhile, the reduction of oxidative phosphorylation is most prominent in the other pole of the principal molecular component. Their progressions altered significantly from week 16 in GK islets with respect to WST islets (Fig 4A, 4E and 4F). Notably, the insulin reduction is slightly later than the alteration of α/β-cell distribution (Fig 7C) while is more drastic after week 16.

The severity of T2D in terms of function includes insulin resistance and islet failure, which are usually measured by glucose levels, insulin levels, glucose tolerance test, etc. In this report, we also quantify the islet deterioration by the α/β-cell distribution. However, only partial functional or morphological data were publicly available in each omic study of T2D.

In the case of GK/WST rats, the insulin levels and the islet micrographs of glucagon and insulin immunohisto-staining were available. The synchronous deteriorations of islet morphological index and transcriptional angiogenesis, as well as those of oxidative phosphorylation and insulin levels shown in the Results section is direct evidence of association between the up/down-regulation of angiogenesis/oxidative phosphorylation pathways and T2D severity. Moreover, age is a fair overall index of T2D severity in the GK rat model. The association between age and the up/down-regulation of angiogenesis/oxidative phosphorylation pathways is demonstrated in Figs 2A and 4A.

In the human case, the HbA$_{1c}$ (glycosylated hemoglobin) indices of the cadaver donors were provided along with their islet transcriptome. The HbA$_{1c}$ index measures the average blood sugar levels over the past 3 months in an individual, thereby acting as a robust indicator of the T2D severity. In Figs 2B and 3C, samples with higher HbA$_{1c}$ levels tend to display larger loadings. Due to the structure of the dual eigen-analysis (Fig 1B), an islet with larger positive loadings was subject to a more pronounced up/down-regulation of the angiogenesis/oxidative phosphorylation pathways. Taken together, during T2D progression, a sample with elevated HbA$_{1c}$ levels is associated with alterations of the angiogenesis/oxidative phosphorylation pathways in the islet.

## Discussion

Whole transcriptome analysis is a powerful tool for understanding the molecular mechanism of T2D development. However, the treatment-control human islets as those in the GK/WST study are not available. The rationale of developing T2D animal models is that they share certain common molecular mechanism with human. Indeed, the data integration of the GK/WST and human islet data demonstrated that up-regulated abnormal angiogenesis and down-regulated oxidative phosphorylation is shared by both species. The polarized SVD stratified each expression profile into two principal components, one for the sample-specific character, and the other for the shared signature.

A key idea of data integration is the polarized version of singular value decomposition. After polarization, if a principal sample-eigenvectors shows a significant contrast, then the outcomes of association or enrichment analysis of the dual gene-eigenvector are readily interpreted into biology. In the case of designed experiments, the dual contrasts can be verified directly by two-sample comparisons. In the case of observational data, the stability of the gene-eigenvectors is evaluated by re-sampling the islets. The statistical significance of the shared gene-eigenvector is examined by the Robust Rank Aggregation.

Alternatively, we can integrate the two expression profiles by directly concatenating them. Then the dual eigen-analysis is applied to the concatenated matrix. After both expression profiles are normalized so that the modes of gene differentiations between pairwise samples are near zero, the differentiation of the house-keeping genes from any contrast is guaranteed to be insignificant. Consequently, the remaining significant gene differentiation in the principal eigenvector would correspond to true biology. The shared biology, if any, between the two expression profiles would be captured by the principal gene eigenvector. The results can be verified by examining the association between the polarized loadings of each species from the integrated sample eigenvector and the respective T2D factors (Fig 3B and 3C). The results can further be validated by examining the consistency between the enrichment outcomes of the species-alone and integrated gene eigenvectors (Figs 2C and 3D).

Moreover, the integration results can be extended to other expression profile data when they become available. Thus, the current outcomes can be either strengthened or falsified by further experiments/data on the same platform.

In the model of ZDF rats, increased expression of several hypoxia- and angiogenesis-related genes was reported, *c.f.* Fig 1 in [3]. Here we provided, in GK rats and human, a comprehensive picture of up-regulated genes related with angiogenesis and response to hypoxia, including a large portion of genes involved in HIF signaling pathway.

Under normoxia, HIF-1α protein would be degraded through ubiquitination. VHL and other genes involved in its ubiquitination were not up-regulated as observed in the aggregated gene-eigenvector. Under hypoxia, HIF-1α interacts with other co-factors and modulates the transcription of genes that increase oxygen delivery and mediate adaptive responses to oxygen deprivation. The adaptation includes angiogenesis and promoting anaerobic metabolism. Some up-regulated genes downstream HIF signaling pathway were shown in Fig 5C. Among them, the up-regulation of *ALDOB* was confirmed by RT-PCR in GK rats (S5 Fig). Besides, the GO biological process "response to hypoxia" is enriched (S7 Table). Consistently, the islet blood flow in GK rats pronouncedly dropped after 12 weeks [37]. These pieces of evidence suggest that HIF signaling pathway functions under hypoxic stress along T2D progression.

What was reported in this report is about the angiogenesis at the transcriptional level. Although it does not equate to alteration of the islet vascular morphology, they are closely related. On one hand, in human type 2 diabetic islets, increased capillary density, thickened and fragmented capillaries were observed [4]. On the other hand, apart from the pro-angiogenic genes, anti-angiogenic genes including *THBS1*, *THBS2*, *PEDF*, and *DCN* were found at the top of the high-ranking list (Fig 5B). Their functions are helpful for understanding the above phenomena observed on islet blood vessels. The role of THBS1 has been implicated in diabetes [46]. Furthermore, PEDF selectively destroys newly forming vessels but does not appear to harm the existing one. Its high expression provides a clue to the alteration of the vascular morphology.

Corresponding to the transcriptional angiogenesis at the molecular level, we consider the alteration of α/β-cell distribution at the morphological level for the following reasons. First, the distribution of α- and β-cells changed from a core-mantle organization in normal rodent islets [8], to a more mixing distribution [9–11] in diabetic islets. On the other hand, during pancreas development overexpression of VEGF-A that increased vascular endothelial cell number would lead to scattered α- and β-cells that never coalesced into islets [7]. Second, in normal islets, α-cells are in direct contact with vasculature [50]. It is then likely that the change in the vasculature might be accompanied with the change in distribution of α-cells. Third, from the insulin and glucagon distribution we defined a morphometric index to quantify the α/β-cell distribution irregularity. The alterations of the α/β-cell distribution at the morphological and abnormal angiogenesis at the molecular level turned out to be synchronous. To our best knowledge, this is the first report on the synchronous alterations of the micro-level angiogenesis and the macro-level morphology. Yet the mechanism that connects them is still missing.

The functional alterations in terms of insulin levels and the molecular alterations of oxidative phosphorylation are synchronous as well. More specifically, the alteration of insulin levels of GK rats before and after week 8, which is not simply monotone, corresponds to the top two expression eigenvectors. The downtrend after week 8 corresponded to the first eigenvector characterized by increased angiogenesis and reduced oxidative phosphorylation, while the compensatory uptrend during week 4 to 8, still in the development stage, corresponded to a different mechanism manifested in the second eigenvector. Around week 8 in GK rats, the heightened *INS1* mRNA level (Fig 4E) is well consistent with the plasma hyperinsulinemia

[36] (S9 Fig). The change in oxidative phosphorylation is in line with that of the islet blood flow in GK rats.

We note that the alteration of α/β-cell distribution which is synchronous to that of transcriptional angiogenesis occurs earlier than the insulin deterioration in GK rats (Fig 7C). Specifically, at week 8, when the insulin level is heightened, morphological alteration of islet already occurs. This observation supports the view raised in our previous review article [51] that screening for prediabetes should include fasting and postprandial blood insulin levels along with blood glucose levels; Moreover, to prevent the islet morphological alteration from becoming irreversible, or to prevent T2D pathological progression from irreversible, certain pharmacological intervention should be taken at as early as the stage of hyperinsulinemia even with euglycemia.

In the case of GK versus WST rats, the differences at the micro-level such as up-regulated angiogenesis and down-regulated oxidative phosphorylation and those at the macro-level such as alteration of α/β-cell distribution, reduced insulin levels, and reduced blood flow (Fig 7A) are surely caused by the genetic factors. The causal process from the genetic factors to the macro-level alterations is thought to be mediated by the micro-level alterations. The mediation is extended to the human data, which is observational, by integrating the expression profiles of the two species at the micro-level. The extrapolation is to some extent supported by the association between the sample loadings and the HbA$_{1c}$ levels (Figs 2B and 3C). In addition, the HIF-1 signaling pathway, which is upstream of both angiogenesis and oxidative phosphorylation, is up-regulated in both rat and human at the micro-level.

The analysis of the expression profiles of rat and human islets supports the existence of aberrant angiogenesis and its counteractive mechanism at the transcriptional level in the type 2 diabetic islets. However, certain gaps between the alterations of transcriptions and those of islet morphology need to be filled by approaches including molecular knockouts of the key genes reported in this report. Rather than bulk expression profiles of islets, the spatial transcriptome at the single cell level would be of great help in dissecting the details of the abnormal angiogenesis.

## Supporting information

**S1 Text. Advantages of dual eigen-analysis.**
(DOCX)

**S2 Text. The three types of angioackngenic genes significantly up-regulated in the aggregated gene-eigenvector of rat and human.**
(DOCX)

**S3 Text. Metabolism switch in the gene signature.**
(DOCX)

**S4 Text. Protective mechanisms of decorin and PEDF.**
(DOCX)

**S1 Fig. Densities of pairwise differences between rat sample 1 and sample 2 before and after normalization by MUREN.**
(PDF)

**S2 Fig. Density of pairwise differences between human samples GSM946735 and GSM946750 after sub-sub normalization.**
(PDF)

**S3 Fig. Sorted loadings of the zeroth principal sample-eigenvector of rat and human.**
(PDF)

**S4 Fig. Sorted loadings of the first principal sample-eigenvector of rat data excluding week 4.**
(PDF)

**S5 Fig. Consistency between the time-course expression of GK and WST islets by RNA-seq and RT-PCR data.**
(PDF)

**S6 Fig. Consistency between the time-course expression of GK and WST islets by RNA-seq and RT-PCR data of the 4-month-old rats from another study.**
(PDF)

**S7 Fig. The stability of the human gene-eigenvectors by re-sampling islets.**
(PDF)

**S8 Fig. Quantification of pancreatic α- and β-cells spatial distribution along the T2D progression.**
(PDF)

**S9 Fig. The plasma insulin levels of GK and WKY rats over time.**
(PDF)

**S10 Fig. Sorted first principal sample-eigenvector of human.**
(PDF)

**S1 Table. Top three squared singular values of the two pancreatic islet expression profiles and their percentages.**
(DOCX)

**S2 Table. Significantly up-regulated genes involved in the angiogenesis-related GO biological processes in the aggregated gene-eigenvector.**
(DOCX)

**S3 Table. Significantly up-regulated endothelial cell marker genes in the aggregated gene-eigenvector.**
(DOCX)

**S4 Table. Significantly up-regulated genes involved in the KEGG "VEGF signaling pathway" in the aggregated gene-eigenvector.**
(DOCX)

**S5 Table. Significantly up-regulated genes involved in the REACTOME pathway "signaling by VEGF" in the aggregated gene-eigenvector.**
(DOCX)

**S6 Table. Significantly up-regulated genes involved in the KEGG "HIF-1 signaling pathway" in the aggregated gene-eigenvector.**
(DOCX)

**S7 Table. Significantly up-regulated genes related to hypoxia in the aggregated gene-eigenvector.**
(DOCX)

**S8 Table. Inflammation-related pathways enriched at the positive poles of the angiogenesis gene-eigenvectors of the two species.**
(DOCX)

**S9 Table. Inflammation-related pathways enriched at the positive poles of the angiogenesis gene-eigenvectors of the two species.**
(DOCX)

**S10 Table. P-values of angiogenesis-related pathways evidenced by enrichment analysis of human's first principal gene-eigenvector.**
(DOCX)

## Acknowledgments

The authors thank Jianhui Shi affiliated with the Academy of Mathematics and Systems Science, Chinese Academy of Sciences, for his guidance on the software usage of gene set enrichment analysis. The authors would like to thank Professor Xiaoshan Gao from Academy of Mathematics and Systems Science, Chinese Academy of Sciences for his support of this interdisciplinary research.

## Author Contributions

**Conceptualization:** Lei M. Li.

**Data curation:** Shenghao Cao, Linting Wang, Yance Feng, Lei M. Li.

**Formal analysis:** Shenghao Cao, Linting Wang, Yance Feng, Lei M. Li.

**Funding acquisition:** Lei M. Li.

**Investigation:** Shenghao Cao, Linting Wang, Xiao-ding Peng, Lei M. Li.

**Methodology:** Shenghao Cao, Linting Wang, Lei M. Li.

**Project administration:** Lei M. Li.

**Resources:** Linting Wang, Lei M. Li.

**Software:** Shenghao Cao, Linting Wang, Yance Feng, Lei M. Li.

**Supervision:** Lei M. Li.

**Validation:** Shenghao Cao, Linting Wang, Lei M. Li.

**Visualization:** Shenghao Cao, Linting Wang, Lei M. Li.

**Writing – original draft:** Shenghao Cao, Linting Wang, Lei M. Li.

**Writing – review & editing:** Shenghao Cao, Linting Wang, Xiao-ding Peng, Lei M. Li.

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
