## [Decision Letter · Decision Letter 0]

7 Jun 2023

PONE-D-23-14278A data integration approach unveils a transcriptional signature of type 2 diabetes progression in rat and human isletsPLOS ONE

Dear Dr. Li,

Thank you for submitting your manuscript to PLOS ONE. After careful consideration, we feel that it has merit but does not fully meet PLOS ONE’s publication criteria as it currently stands. Therefore, we invite you to submit a revised version of the manuscript that addresses the points raised during the review process.

We look forward to receiving your revised manuscript.

Kind regards,

Kanhaiya Singh, Ph.D

Academic Editor

PLOS ONE

Journal Requirements:

Additional Editor Comments:

Please respond to the comments raised by reviewer about integrating human and mouse in the SVD step.

Reviewers' comments:

Reviewer's Responses to Questions

**Comments to the Author**

1. Is the manuscript technically sound, and do the data support the conclusions?

Reviewer #1: Yes

Reviewer #2: Yes

2. Has the statistical analysis been performed appropriately and rigorously? 

Reviewer #1: Yes

Reviewer #2: Yes

3. Have the authors made all data underlying the findings in their manuscript fully available?

Reviewer #1: Yes

Reviewer #2: Yes

4. Is the manuscript presented in an intelligible fashion and written in standard English?

Reviewer #1: Yes

Reviewer #2: Yes

5. Review Comments to the Author

Reviewer #1: In this manuscript, Cao et al. developed a computational method to integrate and compare the transcriptomic data for human and rat islets. The method is based on SVD followed by dual-eigenvalue analysis of gene expression data from a human observational data and mouse islet data. Their analysis identified a conserved expression pattern across the two species as manifested by a principal gene-eigenvector characterized by up-regulated angiogenesis and down-regulated oxidative phosphorylation. The method and results from the selected human/mouse data provide comprehensive and interesting insights on the de-regulated pathways underlying the T2D diseases. The manuscript is clearly written with detailed description about the key pathways identified from the bioinformatics analysis.

Major comments:

1. The SVD analysis was performed separately for human and mouse data, and the comparation between the two species was focused on the pathways enriched in the top gene eigenvectors and the association between sample loading (in the top sample eigenvectors). Is there anyway to match eigen-vectors (both sample and gene) between human and mouse? Is there any way to integrate the human and mouse in the SVD step? It will be great if these problems can be discussed.

2. Whether the identified pathways are associated with T2D disease severity? There might be some T2D data that can be used to further test this.

Reviewer #2: Journal: PLOS ONE

Manuscript Type: Research Article

Manuscript ID: PONE-D-23-14278

Title: A data integration approach unveils a transcriptional signature of type 2 diabetes progression in rat and human islets

The current work by Cao et al investigates molecular insights into the pathology of islets in type 2 diabetes. Authors developed a computational approach to integrating expression profiles of Goto-Kakizaki and Wistar rat islets from a designed experiment with those of the human islets from an observational study. Overall evaluation shows that the study has been executed well. There are few minor suggestions. If rectified, the article will be suitable for publication in PlosOne.

1. The authors should elaborate the introduction and discussion sections.

2. The limitations and prospects of the study should be highlighted in the discussion section.

6. PLOS authors have the option to publish the peer review history of their article (what does this mean?). If published, this will include your full peer review and any attached files.

Reviewer #1: No

Reviewer #2: No

---

## [Author Response · Author response to Decision Letter 0]

10 Jul 2023

We are very grateful to the reviewers' insightful comments and suggestions. We have made an endeavor to improve the research and to revise the manuscript according to their suggestions. The point-to-point responses to the editors' and reviewers' comments are as follows. 

Additional Editor Comments:

Please respond to the comments raised by reviewer about integrating human and mouse in the SVD step.

Response: 

As suggested by Reviewer 1, we included an alternative direct integration. That is, after baseline removal and variation equalization, we directly concatenated the two expression profiles, aligning their homologous genes to form a single matrix. Next, we applied the unified SVD procedure to the concatenated expression matrix. The detailed method is described in the new subsection "Unified SVD of the concatenated expression profile matrix of two species". 

The detailed results are described in the new subsection "Similar principal eigen-component observed in the unified SVD of the concatenated expression profile". The method and results of the direct integration by SVD are summarized in the new Fig 3.

The code that includes the implementation of the unified SVD was updated at https://codeocean.com/ with the DOI: https://doi.org/10.24433/CO.1261622.v3

Reviewer: 1

Comments to the Author

In this manuscript, Cao et al. developed a computational method to integrate and compare the transcriptomic data for human and rat islets. The method is based on SVD followed by dual-eigenvalue analysis of gene expression data from a human observational data and mouse islet data. Their analysis identified a conserved expression pattern across the two species as manifested by a principal gene-eigenvector characterized by up-regulated angiogenesis and down-regulated oxidative phosphorylation. The method and results from the selected human/mouse data provide comprehensive and interesting insights on the de-regulated pathways underlying the T2D diseases. The manuscript is clearly written with detailed description about the key pathways identified from the bioinformatics analysis.

Response: Thanks for your appreciation and the comments. Indeed, we tried to gain fresh insights into the molecular mechanisms underlying the type 2 diabetes progression by developing new computational methods that can integrate the two comprehensive islet transcriptomic data, one from a designed experiments of GK/WST rats and the other from an observational study of human. The work aims at providing valuable guidance in the prevention and treatment of T2D.

1) The SVD analysis was performed separately for human and mouse data, and the comparation between the two species was focused on the pathways enriched in the top gene eigenvectors and the association between sample loading (in the top sample eigenvectors). Is there any way to match eigen-vectors (both sample and gene) between human and mouse? Is there any way to integrate the human and mouse in the SVD step? It will be great if these problems can be discussed.

Response: This is a very good point regarding data integration by SVD and dual eigen-analysis. First, we would like to make a clarification. Although in an earlier report [1] we showed angiogenesis was up-regulated in the mRNA expressions of whole pancreas from outbred mouse fed with high fat diet, in the present article we investigated the expression profiles of GK/WST rat and human islets. The GK rat is one of the best-characterized spontaneous T2D animal models, and its disease progression is more homogeneous over time among individuals than the outbred mouse model.

In the last submission, we integrated the expression profiles of the rat and human islets by carrying out SVD on each of the two matrices and then matching their gene-eigenvectors by the Robust Rank Aggregation. Now in the revision, we included an alternative direct integration. That is, after baseline removal and variation equalization, we directly concatenated the two expression profiles, aligning their homologous genes to form a single matrix. Next, we applied the SVD and dual eigen-analysis procedure to the concatenated expression matrix. The detailed method is described in the new Materials and Methods subsection "Unified SVD of the concatenated expression profile matrix of two species".

On one hand, in the first sample-eigenvector of the concatenated expression matrix, the positive and negative loadings of samples from both rat and human correspond to T2D and normal samples roughly and respectively. Furthermore, the first sample-eigenvector can be partitioned into the rat and human components. In the rat component, a significant contrast was observed between GK and WST sample loadings. In the human component, a significant contrast was observed between high- and low-HbA1c sample loadings. The results were consistent with those obtained from the species-alone SVDs.

On the other hand, the first gene-eigenvector of the concatenated expression matrix matched the first of rat and the second of human as well. The enrichment analysis showed that the angiogenesis pathways were up-regulated while the oxidative phosphorylation pathways were down-regulated. This transcriptional signature was consistent with that in the aggregated gene-eigenvector obtained from the species-alone SVDs.

The detailed results are described in the new Results subsection "Similar principal eigen-component observed in the unified SVD of the concatenated expression profile". The method and results of the direct integration by the unified SVD are summarized in the new Fig 3.

The code that includes the implementation of the unified SVD was updated at https://codeocean.com/ with the DOI: https://doi.org/10.24433/CO.1261622.v3

2) Whether the identified pathways are associated with T2D disease severity? There might be some T2D data that can be used to further test this.

Response: Indeed, one aim of this report is, by integrating the two publicly available data sets, to establish the association between the up/down-regulation of angiogenesis/oxidative phosphorylation pathways at the molecular-level and the severity of T2D progression at the macro-level. Now we elaborated this association at the end of the Results Section.

"The severity of T2D in terms of function includes insulin resistance and islet failure, which are usually measured by glucose levels, insulin levels, glucose tolerance test, etc. In this report, we also quantify the islet deterioration by the α/β-cell distribution. However, only partial functional or morphological data were publicly available in each omic study of T2D. 

In the case of GK/WST rats, the insulin levels and the islet micrographs of glucagon and insulin immunohisto-staining were available. The synchronous deteriorations of islet morphological index and transcriptional angiogenesis, as well as those of oxidative phosphorylation and insulin levels shown in the Results section is direct evidence of association between the up/down-regulation of angiogenesis/oxidative phosphorylation pathways and T2D severity. Moreover, age is a fair overall index of T2D severity in the GK rat model. The association between age and the up/down-regulation of angiogenesis/oxidative phosphorylation pathways is demonstrated in Figs 2A and 4A. 

In the human case, the HbA1c (glycosylated hemoglobin) indices of the cadaver donors were provided along with their islet transcriptome. The HbA1c index measures the average blood sugar levels over the past 3 months in an individual, thereby acting as a robust indicator of the T2D severity. In Figs 2B and 3C, samples with higher HbA1c levels tend to display larger loadings. Due to the structure of the dual eigen-analysis (Fig 1B), an islet with larger positive loadings was subject to a more pronounced up/down-regulation of the angiogenesis/oxidative phosphorylation pathways. Taken together, during T2D progression, a sample with elevated HbA1c levels is associated with alterations of the angiogenesis/oxidative phosphorylation pathways in the islet."

Large islet data including both expression profiles and T2D pathological indices are valuable yet rare. As shown in the report, c.f. the 4-th paragraph in the discussion section, as more T2D expression profiles would be available, they could be integrated with the GK/WST rats and human islet data used in this study. 

Reviewer: 2

Comments to the Author

The current work by Cao et al investigates molecular insights into the pathology of islets in type 2 diabetes. Authors developed a computational approach to integrating expression profiles of Goto-Kakizaki and Wistar rat islets from a designed experiment with those of the human islets from an observational study. Overall evaluation shows that the study has been executed well. There are few minor suggestions. If rectified, the article will be suitable for publication in PlosOne.

Response: Thanks for your appreciation and comments. We have tried to improve the manuscript according to your suggestions.

1) The authors should elaborate the introduction and discussion sections.

Response: As suggested, we elaborated the introduction and discussion sections in the revision. The new contents are highlighted in these two sections.

In particular, in the introduction section, we added some detail of the abnormal islet capillary morphology reported in the literature. 

In the discussion section, we elaborated on the direct concatenation of the two expression profiles followed by a unified SVD. This unified SVD approach is an alternative integration that was added in the current revision. The second paragraph from the end discussed the reasoning of integrated causal inference in the study. 

"In the case of GK versus WST rats, the differences at the micro-level such as up-regulated angiogenesis and down-regulated oxidative phosphorylation and those at the macro-level such as alteration of α/β-cell distribution, reduced insulin levels, and reduced blood flow (Fig 7A) are surely caused by the genetic factors. The causal process from the genetic factors to the macro-level alterations is thought to be mediated by the micro-level alterations. The mediation is extended to the human data, which is observational, by integrating the expression profiles of the two species at the micro-level. The extrapolation is to some extent supported by the association between the sample loadings and the HbA1c levels (Figs 2B and 3C). In addition, the HIF-1 signaling pathway, which is upstream of both angiogenesis and oxidative phosphorylation, is up-regulated in both rats and human at the micro-level."

2) The limitations and prospects of the study should be highlighted in the discussion section.

Response: Yes, we highlighted the limitations of the study. In the discussion section, at the end of the 8-th paragraph, we added,

“To our best knowledge, this is the first report on the synchronous alterations of the micro-level angiogenesis and the macro-level morphology. Yet the mechanism that connects them is still missing.”

At the end of the discussion, we pointed out the limitation of the bulk sequencing data and the future work and prospect of spatial single cell transcriptome.

“However, certain gaps between the alterations of transcriptions and those of islet morphology need to be filled by approaches including molecular knockouts of the key genes reported in this report. Rather than bulk expression profiles of islets, the spatial transcriptome at the single cell level would be of great help in dissecting the details of the abnormal angiogenesis.”

We also added the clinical implications of this study in the discussion section, the third paragraph from the end, as follows.

"We note that the alteration of α/β-cell distribution which is synchronous to that of transcriptional angiogenesis occurs earlier than the insulin deterioration in GK rats (Fig 7C). Specifically, at week 8, when the insulin level is heightened, morphological alteration of islet already occurs. This observation supports the view raised in our previous review article [2] that screening for prediabetes should include fasting and postprandial blood insulin levels along with blood glucose levels; Moreover, to prevent the islet morphological alteration from becoming irreversible, or to prevent T2D pathological progression from irreversible, certain pharmacological intervention should be taken at as early as the stage of hyperinsulinemia even with euglycemia."

References

1. Li LM, Liu XX, Wang L, Wang Y, Tian X, Gong F, et al. A Novel Dual Eigen-Analysis of Mouse Multi-Tissues' Expression Profiles Unveils New Perspectives into Type 2 Diabetes. Sci Rep. 2017;7(1):5044.

2. Peng X-D, Li LM. A New Angle in Tackling the Problems of Curing Type 2 Diabetes: Detect it Earlier, and Treat it Earlier. Medical Recapitulate. 2018;24(20):4060-7.

---

## [Decision Letter · Decision Letter 1]

25 Sep 2023

A data integration approach unveils a transcriptional signature of type 2 diabetes progression in rat and human islets

PONE-D-23-14278R1

Dear Dr. Li,

We’re pleased to inform you that your manuscript has been judged scientifically suitable for publication and will be formally accepted for publication once it meets all outstanding technical requirements.

Kind regards,

Kanhaiya Singh, Ph.D

Academic Editor

PLOS ONE

Additional Editor Comments (optional):

Reviewers' comments:

Reviewer's Responses to Questions

**Comments to the Author**

1. If the authors have adequately addressed your comments raised in a previous round of review and you feel that this manuscript is now acceptable for publication, you may indicate that here to bypass the “Comments to the Author” section, enter your conflict of interest statement in the “Confidential to Editor” section, and submit your "Accept" recommendation.

Reviewer #1: All comments have been addressed

2. Is the manuscript technically sound, and do the data support the conclusions?

Reviewer #1: Yes

3. Has the statistical analysis been performed appropriately and rigorously? 

Reviewer #1: Yes

4. Have the authors made all data underlying the findings in their manuscript fully available?

Reviewer #1: Yes

5. Is the manuscript presented in an intelligible fashion and written in standard English?

Reviewer #1: Yes

6. Review Comments to the Author

Reviewer #1: All my comments have been fully addressed. I thank the authors for make the revisions according to my comments.

7. PLOS authors have the option to publish the peer review history of their article (what does this mean?). If published, this will include your full peer review and any attached files.

Reviewer #1: No

---

## [Editor Report · Acceptance letter]

2 Oct 2023

PONE-D-23-14278R1 

A data integration approach unveils a transcriptional signature of type 2 diabetes progression in rat and human islets 

Dear Dr. Li:

I'm pleased to inform you that your manuscript has been deemed suitable for publication in PLOS ONE. Congratulations! Your manuscript is now with our production department. 

Kind regards, 

on behalf of

Dr. Kanhaiya Singh 

Academic Editor

PLOS ONE